# Conformational dynamics and putative substrate extrusion pathways of the *N*-glycosylated outer membrane factor CmeC from *Campylobacter jejuni*

**Kahlan E. Newman**[1], **Syma Khalid**[1,2]*

**1** School of Chemistry, University of Southampton, Southampton, United Kingdom, **2** Department of Biochemistry, University of Oxford, Oxford, United Kingdom

* syma.khalid@bioch.ox.ac.uk

**Data Availability Statement:** All input files necessary to run the simulations, and all resulting trajectories discussed in the paper are available on Zenodo: https://doi.org/10.5281/zenodo.7041473.

## Abstract

The outer membrane factor CmeC of the efflux machinery CmeABC plays an important role in conferring antibiotic and bile resistance to *Campylobacter jejuni*. Curiously, the protein is *N*-glycosylated, with the glycans playing a key role in the effective function of this system. In this work we have employed atomistic equilibrium molecular dynamics simulations of CmeC in a representative model of the *C. jejuni* outer membrane to characterise the dynamics of the protein and its associated glycans. We show that the glycans are more conformationally labile than had previously been thought. The extracellular loops of CmeC visit the open and closed states freely suggesting the absence of a gating mechanism on this side, while the narrow periplasmic entrance remains tightly closed, regulated *via* coordination to solvated cations. We identify several cation binding sites on the interior surface of the protein. Additionally, we used steered molecular dynamics simulations to elucidate translocation pathways for a bile acid and a macrolide antibiotic. These, and additional equilibrium simulations suggest that the anionic bile acid utilises multivalent cations to climb the ladder of acidic residues that line the interior surface of the protein.

## Author summary

*Campylobacter jejuni* is a Gram-negative bacterium that is a major cause of gastroenteritis. Infection is also associated with the development of some auto-immune conditions. Therefore, it is of key biomedical importance.

Bacterial efflux pumps extrude waste and harmful chemicals, including antibiotics from inside the cell to the extracellular environment, thus providing an impediment to antibacterial treatments. The major efflux system of *C. jejuni* is the cell envelope spanning CmeABC protein complex. The outer membrane factor CmeC plays an important role in conferring antibiotic and bile resistance to *C. jejuni*. We have used molecular dynamics simulations of CmeC to characterise the pathways *via* which a bile acid and an antibiotic move through this glycosylated protein. We identify several cation-binding sites within

**Funding:** K.E.N. is supported by a Ph.D. Studentship from the Engineering and Physical Sciences Research Council (Project No. 2446840). S.K. is supported by an EPSRC Established Career Fellowship (EPSRC grant no. EP/V030779/1). The funders had no role in study design, data collection and analysis, decision to publish, or preparation of the manuscript.

**Competing interests:** The authors have declared that no competing interests exist.

the lumen of the protein and show that the anionic bile acid utilises multivalent cations to climb the ladder of acidic residues that line the interior surface of the protein.

## Introduction

The Gram-negative bacterium *Campylobacter jejuni* is considered a leading cause of gastroenteritis, infecting an estimated 400 million people annually [1]. This thermophilic, microaerophilic enteropathogen is an asymptomatic commensal organism in avian species [2–5]. As such, up to 90% of retailed chicken carcasses are contaminated with *C. jejuni* [6,7] and the leading cause of human infection in developed countries is through inappropriate handling or undercooking of poultry products [2,5,8–12]. Human disease is most commonly acute but self-limiting, presenting as a fever, abdominal cramping and diarrhoea [13,14]. However, infection is also associated with the development of debilitating auto-immune conditions such as Miller-Fisher syndrome and Guillain-Barré syndrome [15–20].

   *C. jejuni* displays resistance to a variety of antibiotic classes through several resistance mechanisms [21,22]. Almost invariably, the main multi-drug efflux system in this bacterium, CmeABC, works synergistically with other mechanisms (such as decreasing outer membrane permeability) [21] to confer a higher level of both intrinsic and acquired resistance to these antimicrobials by reducing their intracellular concentration. In severe cases of bacterial diarrhoea, antibiotics may be prescribed. The macrolide antibiotic erythromycin is used if campylobacteriosis is suspected [23] but fluoroquinolones are commonly used if the infection source is unclear; CmeABC has been shown to contribute resistance to both macrolides and fluoroquinolones, with resistance to the latter class now widespread due to its indiscriminate use in poultry farming and clinical environments [24–26]. CmeABC has also been shown to confer bile resistance [27] which is essential for the survival of this bacterium in its main reservoir of poultry intestinal tracts [2]. This efflux assembly comprises three proteins: the resistance-nodulation-division (RND) inner membrane transporter, CmeB; the periplasmic adaptor protein (PAP), CmeA; and the outer membrane factor (OMF), CmeC.

   The OMF of this machinery is similar in structure to its homologues TolC and CusC (*Escherichia coli*) [28,29], MtrE (*Neisseria gonorrhoeae*) [30], and OprM (*Pseudomonas aeruginosa*) [31]. CmeC is a 'cannon-shaped' homotrimer that forms a 130 Å channel through which substrates can be extruded into the extracellular environment [32]. Three key subdomains of the protein delimit this channel: the β-barrel; the α-barrel; and the coiled coil domains (**Fig 1A**). The β-barrel is ~30 Å tall and spans most of the outer membrane, while the α-barrel and coiled coil domains extend ~100 Å into the periplasmic space where this protein couples to CmeA and presumably the peptidoglycan cell wall. Like TolC, OprM, and CusC, the interior surface of CmeC is highly electronegative with aspartate and glutamate residues constituting a considerable proportion of the lining residues [32] (**Fig 1B**). Each member of the CmeABC assembly is *N*-glycosylated by the attachment of the following heptasaccharide [33] (**S1 Fig**):

   GalNAc-α1,4-GalNAc-α1,4-[Glc-β-1,3]GalNAc-α1,4-GalNAc-α1,4-GalNAc-α1,3-diNAc-Bac-β1 where diNAcBac is N',N'-diacetylbacillosamine [2,4-diacetamido-2,4,6 trideoxyglucopyranose]; GalNAc is N-acetylgalactosamine; and Glc is glucose. Removal of the *N*-glycosylation pathway results in diminished activity of this efflux machinery [34], as well as pleiotropic effects such as reduced motility [35] and virulence [36]. CmeC has two experimentally validated glycosylation sites: $_{11}$EA**N**YS$_{15}$ and $_{28}$ET**N**SS$_{32}$ (residue numbers are *after* signal peptide cleavage).

   Previous *in silico* and mutational studies aiming to elucidate OMF dynamics have primarily focused on TolC from the archetypal Gram-negative bacterium *E. coli* [37–44], though OprM

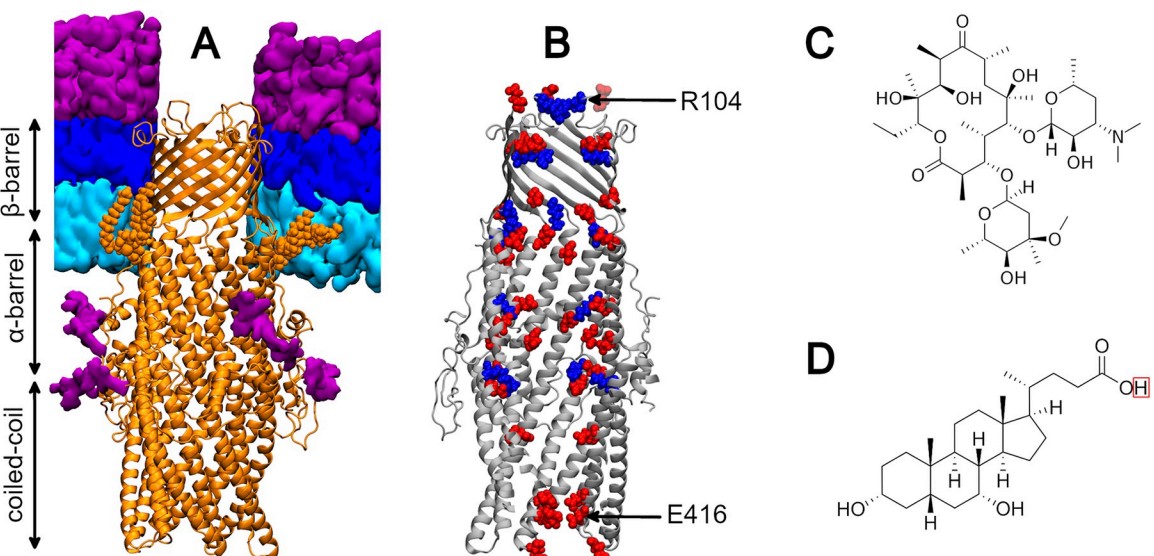

**Fig 1. Protein and substrate structures. (A)** CmeC embedded in a model C. jejuni outer membrane. CmeC shown in orange New Cartoon representation, with N-terminal lipidation shown in orange van der Waals (VDW); phospholipids shown as cyan surface; Lipid A moieties of LOS shown as a dark blue surface; and LOS oligosaccharides and N-glycans shown as a purple surface. The β-barrel and top of the α-barrel are embedded in the hydrophobic membrane core. **(B)** Cutaway of the x-ray crystallographic structure of CmeC (grey New Cartoon), with acidic and basic residues that line the central channel shown in red and blue VDW representation respectively. Residues identified as constrictions in the X-ray crystal structure (R104 and E416) labelled. **(C)** Structure of the macrolide antibiotic erythromycin. **(D)** Structure of the bile acid chenodeoxycholic acid (CDCA). The proton highlighted by the red box is removed in solution to yield a carboxylate group.

from *P. aeruginosa*, and MtrE from *N. gonorrhoeae* have also been simulated [45–47]. These studies have shown several common features of OMFs. These proteins display two constriction regions: one at the periplasmic entrance and one at the extracellular exit. Contrary to initial molecular dynamics (MD) studies [38] OMFs are putatively gated at the periplasmic entrance only [41,43,45]. The extracellular loops that define the extracellular constriction have been shown to visit open and closed states freely, while the periplasmic entrance remains locked closed by extensive inter- and intraprotomer hydrogen bonding and salt bridges, with TolC and OprM also showing interactions with cations [41,43–45]. The coiled-coil domain of these proteins opens in an 'iris-like' fashion at the cost of disrupting the circular hydrogen bonding and salt bridge networks in this region; this occurs spontaneously on coupling to the PAP, but requires energetic input for the isolated OMF to prevent the uncontrolled diffusion of solutes [39,43,44].

Analogous simulation studies of the OMF of CmeABC, CmeC, are absent in the literature and to the best of our knowledge the components of the *N*-glycosylated CmeABC system have not previously been simulated. Here we present equilibrium molecular dynamics simulations of the OMF CmeC in a biologically relevant *C. jejuni* outer membrane model. The results of these simulations suggest that *N*-glycans are more conformationally labile than previously thought, and that CmeC behaves similarly to other OMFs with cation-dependent gating at the periplasmic entrance and no gating mechanism at the extracellular exit. In addition, we present steered molecular dynamics simulations in which two known substrates of this efflux machinery, the macrolide antibiotic erythromycin and the bile acid chenodeoxycholic acid (CDCA) (**Fig 1C and 1D**), are pulled through this outer membrane channel. We investigate the orientation preference of these molecules within the channel, and propose a mechanism of translocation for the anionic CDCA substrate.

## Results and discussion

### CmeC stability within the membrane

Previous simulation studies of OMFs TolC and OprM used simplified membrane models [37,38,40–42,45]. While useful data may still be extracted from these simulations, phospholipid bilayers are not fully representative of the physicochemical properties of Gram-negative outer membranes; experimental and computational studies have shown components of the outer membrane, such as LPS (lipopolysaccharide), to affect the structure and function of outer membrane proteins [48–52]. Recent *in silico* studies of systems containing efflux machinery have used more complex outer membrane models, though the dynamics of the OMF of these systems have not been the main focus of the studies [46,53]. In this work, the *N*-glycosylated CmeC trimer was inserted into a biologically-relevant *C. jejuni* outer membrane model containing ganglioside-mimicry LOS (lipooligosaccharide) in the outer leaflet and a mixture of POPE and POPG (18:1:1 1-palmitoyl-2-oleoyl-sn-glycero-3-phosphoethanolamine and 1-Palmitoyl-2-oleoyl-sn-glycero-3-phosphoglycerol, respectively) in the periplasmic leaflet [54,55]. We note that while this membrane model is closer to being biologically accurate than a simple, symmetrical phospholipid bilayer, lysophospholipids, which have recently been discovered to constitute a large proportion of the *C. jejuni* lipodome [56], are not included.

Three x 500 ns equilibrium simulations were performed to assess the stability of this system. Dimensions of the simulation box plateaued after ~300 ns in all three replicates (**S2 Fig**). The root-mean-square deviation (RMSD) of the protein backbone continues to increase across the simulation in all cases (**S3 Fig**) indicating that CmeC is not yet fully equilibrated. This is to be expected, given the size and complexity of the system and the starting conformation being close to the crystal structure. One simulation was extended to 1,250 ns; the backbone RMSD continued to increase across this longer simulation (**S3 Fig**). The continued rise in RMSD is primarily driven by the β-barrel; this is the domain in which the crystallisation and simulation environments are most different, and full equilibration in this region will be slow due to the use of slow-moving LOS [57] in the outer leaflet. The 3 x 500ns simulation trajectories were concatenated and principal component analysis (PCA) was undertaken. Each of the trajectories was projected along the first three eigenvectors (which cumulatively accounted for 32.3% of the variance, **S1 Table**). Each trajectory sampled different subspaces unique from the crystal structure (**Fig 2A**), therefore in combination these replicates likely give better coverage of the conformational landscape than we would expect from a single trajectory of the same length.

Secondary structure analysis (SSA) of the protein shows CmeC to be stable in the membrane, with no substantial changes in secondary structure over the course of the simulations (**S4 Fig**). The root-mean-square fluctuation (RMSF) is consistent with the SSA and with the topology defined by Su *et al.* from the crystal structure [32]; the greatest fluctuations are observed for residues in unstructured regions such as the first extracellular loop (residues 96–108, henceforth referred to as L1), N- and C-termini, and between α-helices at the periplasmic end of the protein (e.g. residues 195–200) (**Fig 2B**). The N-terminus is lipidated, anchored to the inner leaflet of the membrane, hence it displays lower RMSF values than the C-terminus. Interestingly, the second extracellular loop (residues 309–323, henceforth referred to as L2) shows substantially less fluctuation than the first loop. The L1 loops are directed upwards, normal to the membrane or tilted towards the centre of the β-barrel, whereas the L2 loops are parallel to the membrane, projecting out into the LOS headgroups (**Fig 3A and 3B**).

Hydrogen bond analysis (cut-offs of 3 Å and 20°) reveals multiple high-occupancy interactions between LOS and residues within L2. The basic side chains of residues K322 and K318 interact extensively with the phosphate groups of the LOS molecules (**Fig 3C**): hydrogen

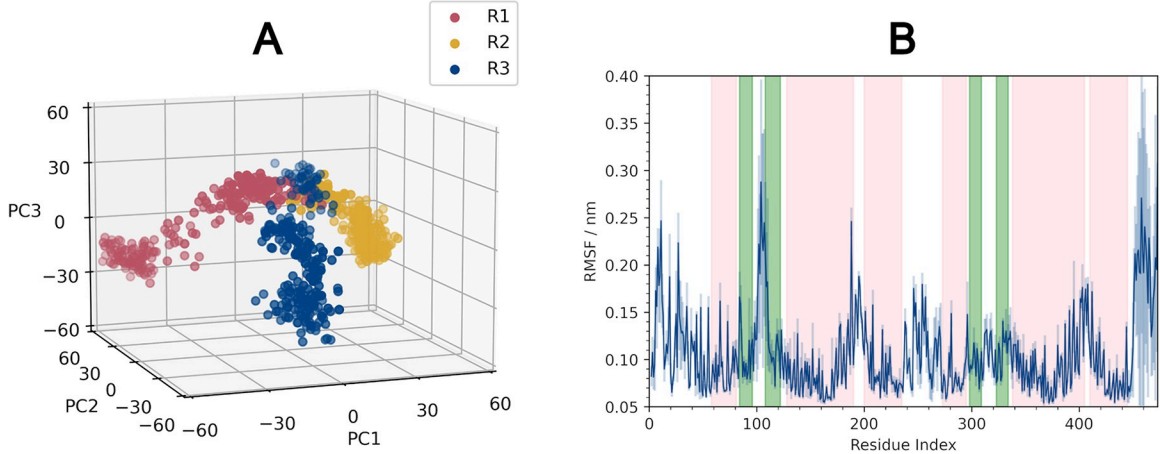

**Fig 2. Analysis of the protein dynamics. (A)** Projections of the backbone trajectories of each replicate along the first three eigenvectors. Each replicate samples a different conformational space, each unique from the crystal structure. **(B)** Root-mean-square fluctuation (RMSF) by residue (calculated from the protein backbone, averaged across the three protomers in each of the three replicates). Average trace shown as a solid blue line, associated standard deviation shown in pale blue. Background coloured by secondary structure: pink = α-helices, green = β-sheets, white = no formally defined secondary structure.

bonds between the lysine side chains and LOS phosphates were observed in >80% of trajectory frames. Acidic residues D312 and D314 were observed to hydrogen bond to core sugar moieties KDO and heptose of nearby LOS molecules. The occupancy of these hydrogen bonds was much lower, at an average of 11% of frames; the sugar moieties are further from the L2 residues and more dynamic than the phosphate head groups, so these interactions are less prevalent. In addition, these acidic residues were observed to coordinate calcium ions (**Fig 3D**) found in the outer leaflet. It is of note that the above interactions would not have been observed had we utilised a simplified phospholipid membrane model, and similar interactions were found when OprM was modelled in a *P. aeruginosa* outer membrane [46]. Finally, the hydrophobic side chain of L313 was observed to embed itself amongst the acyl tails of the LOS molecules in the hydrophobic membrane core, anchoring the L2 loops into the outer leaflet (**Fig 3D**).

## *N*-Glycan dynamics & interactions

A previous MD study on the effect of *N*-glycans on the global structure and dynamics of non-membrane proteins indicated that glycans reduce the dynamic fluctuations of the proteins, enhancing their stability without significantly affecting local or global protein structure [58]. The effect of *N*-glycosylation on CmeC (and the other components of this efflux machinery) will be the focus of later work and so comparison to unglycosylated CmeC is not discussed here. We will, however, discuss the dynamics of these heptasaccharides and the interactions with species within this system.

We observe the glycans to adopt a variety of orientations relative to the principal axis of the protein (approximately parallel to the z-axis and normal to the membrane plane) (**Fig 4A–4C**). This is shown in the timeseries data of the angles (**S5 Fig**) defined between the glycan vectors (defined by Cα of the N13 or N30 residue and the C4 position of the terminal galactosamine, also defining the end-to-end length of the glycan, **Fig 4D**) and the principal axis of the protein. In different relative conformations these sugars can interact with different species within the system. At small angles the glycans point 'upwards' towards the periplasmic leaflet of the outer membrane where they can interact with the unstructured N-terminal residues (the N-terminus is lipidated and anchored to the inner leaflet) and the POPE and POPG

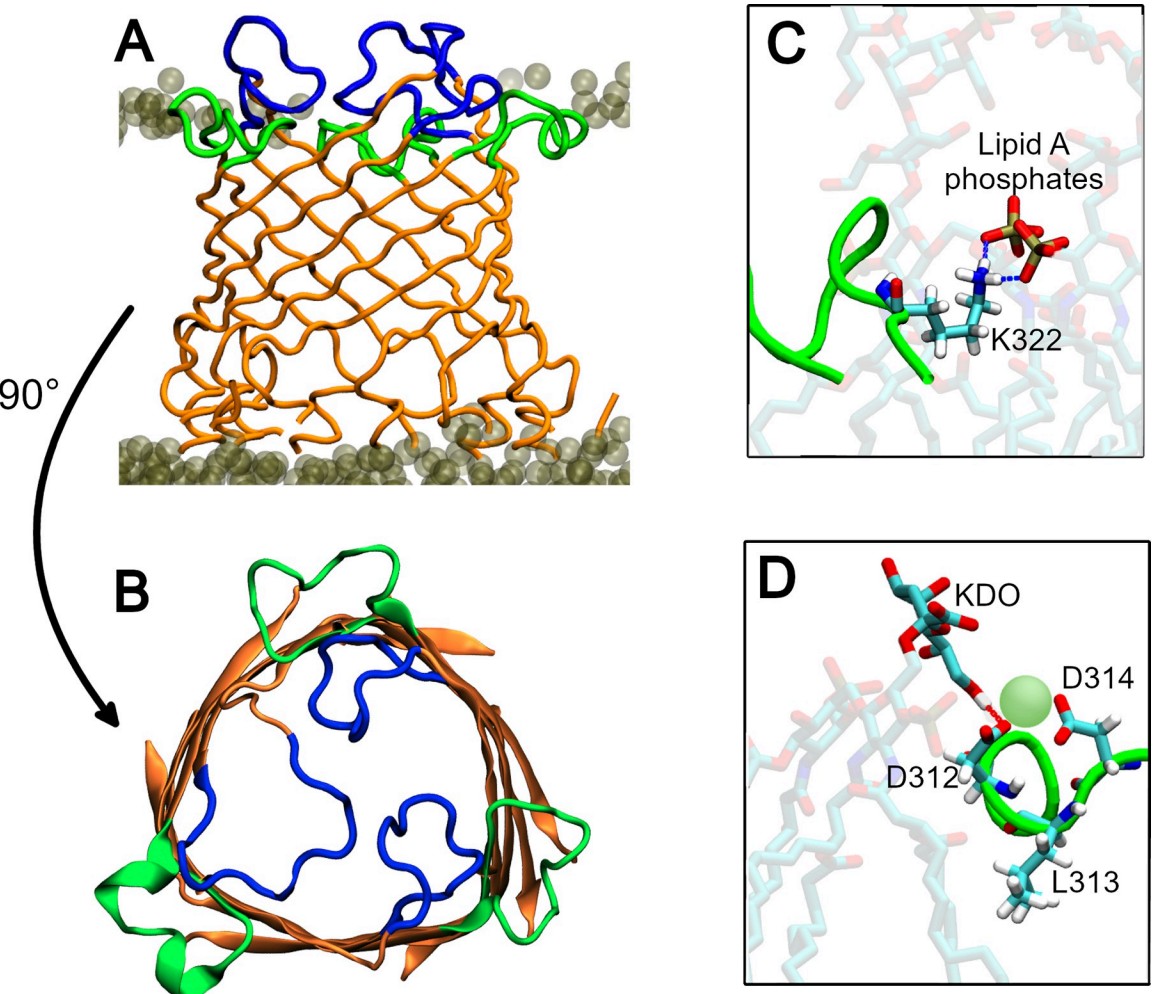

**Fig 3. (A)** Side-view of the CmeC β-barrel in the outer membrane model; **(B)** top-down view of the CmeC β-barrel. β-sheets and top of the α-barrel in orange, residues 96–108 (extracellular loop 1, L1) in blue, residues 309–323 (extracellular loop 2, L2) in green. **(A & B)** show L1 loops to project upwards and into the center of the β-barrel, but L2 loops to project out radially into the outer leaflet of the outer membrane. **(C)** Salt bridge formed between K322 residue in L2 and the phosphates of neighbouring lipid A moieties. **(D)** Acidic residues D312 and D314 can simultaneously coordinate calcium ions (pale green) from amongst the LOS headgroups, and hydrogen bond with KDO sugar moieties in neighbouring LOS molecules. The hydrophobic side chain of L313 is embedded amongst the lipid tails within the membrane core, anchoring L2 to the outer leaflet.

headgroups (**Fig 4A and 4B**). At larger angles the glycans protrude 'downwards' into the periplasmic space. Here the separate heptasaccharide units may hydrogen bond to one another (**Fig 4C**).

The flexibility of the glycans was assessed *via* the end-to-end lengths and the glycosidic torsion angles defined by the linkages between saccharide moieties. The end-to-end length was observed to fluctuate over the course of the simulations, with minimum and maximum lengths of 20.40 and 29.66 Å respectively, and a mean length of 26.84 ± 1.11 Å (standard deviation). A histogram of the end-to-end lengths across the three replicates is shown in **S6 Fig**. This distribution of lengths implies that there is some inherent flexibility in the glycans.

To further investigate this flexibility, the glycosidic torsion angles Ψ and Φ were calculated for each glycosidic linkage for each frame across the three simulations. We define these

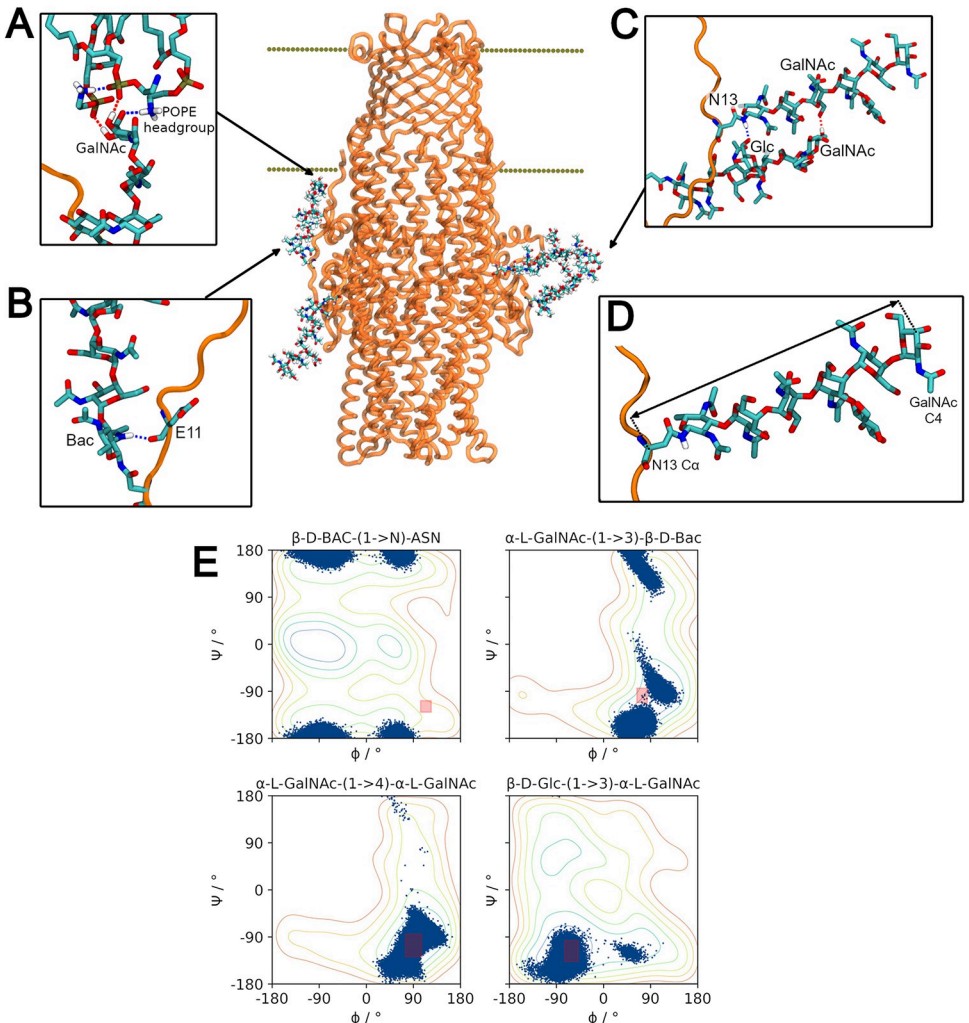

**Fig 4. Glycan dynamics and interactions. (A)** Terminal GalNAc moiety hydrogen bonds to the POPE headgroups in the inner leaflet of the membrane. **(B)** Bacillossamine moiety hydrogen bonds to the backbone of E11 in the unstructured N-terminus region of CmeC. **(C)** Glycans in the same protomer are observed to hydrogen bond to one another. **(D)** Vector defining the end-to-end length of the heptasaccharide and the direction in which the glycan is oriented. Defined between the Cα of the asparagine residue and the C4 position of the terminal GalNAc moiety. **(E)** Scatter plot of the Ψ and Φ torsion angles for each unique glycosidic linkage in the heptasaccharide, plotted in blue atop the free energy contour plots generated via metadynamics simulations by Pedebos et al (Reprinted (adapted) with permission from Ref. 60. Copyright 2015 Oxford University Press). The mean value ± the standard deviation observed for each linkage in Ref. 60 are plotted as pink rectangles on the contour plots. The glycans in our simulations show considerable exploration of the energy landscape.

dihedrals Φ as the angle: O5-C1-O1-CX, and Ψ as the angle: C1-O1-CX-C(X-1). Slynko *et al.* used NMR to assess the glycosidic torsion angles of this heptasaccharide attached to a truncated CmeA monomer, though different definitions to those defined here were used [59]. Across their 20 best structures the glycans occupied a very limited conformational space within minima on the energy surfaces for these linkages [59]. In 2015 Pedebos *et al.* used metadynamics to generate potential energy surfaces defined by the above torsion angles Ψ and Φ for each unique glycosidic linkage in this glycan, and employed MD simulations to investigate the equilibrium values of these linkages in a solvated, peptide-linked heptasaccharide [60]. Similarly, they found the glycans to be rigid with little variation in Ψ and Φ in their 200 ns

simulation. In **Fig 4E** the torsion angles observed over the course of our three equilibrium simulations are plotted over the contour plots produced by Pedebos *et al* [60]: in contrast to both the previous simulation and NMR studies, there is considerable variation in these angles. We observe the glycans to predominantly occupy the global minima on the energy surfaces, exploring the basins on the contour maps. The exception to this is the bacillosamine-asparagine linkage which, in agreement with the molecular dynamics presented by Pedebos *et al*, explores only local minima [60].

Pedebos *et al.* present average values for each linkage with associated standard deviation [60]; whilst the average values we observe for these angles are similar, the SD in these values are considerably greater as the distribution of these values is not always unimodal (**S2 Table and S7 Fig**). Slynko *et al.* present only 20 structures for the glycosidic torsion angles [59], and Pedebos *et al.* present 200 ns of molecular dynamics simulations for a single glycan [60]. The data we present here is collected across 1,500 ns of simulation time (15,000 frames, each with 6 glycans, for a total of 90,000 heptasaccharide conformations). The increased variation in our results compared to previous works is likely due to this increased sampling of the conformational landscape.

## CmeC shape

The β-barrels of OMFs are known to adopt different shapes when simulated in membranes under equilibrium conditions. The β-barrel of TolC has been shown to transition from a cylinder in the crystal structure [28] (PDB ID 1EK9) to a triangular prism [38] and that of OprM is strongly triangular prismatic in its crystal structure [61] (PDB ID 3D5K), which is maintained when simulated [45]. In comparison, CmeC does not appear to adopt a well-defined cylinder or triangular prism during simulations, instead adopting something closer to a hexagonal prism (**Fig 5**). These differences may be explained in part by the positioning of glycine residues; it has recently been shown that inwards-facing glycine residues result in sharp turns/corners in β-barrel membrane proteins [62]. The 'corners' defining the triangular nature of the OprM and TolC β-barrels align with the reasonably well-defined positioning of glycine residue clusters in the x-y plane. By contrast, the glycine residues in CmeC are distributed across 6 less well-defined clusters in the x-y plane of the CmeC β-barrel: one beneath each L1 loop, and one approximately at the centre of each face of the triangle defined by the three L1 loops. This aligns well with the six 'corners' observed in the simulated CmeC.

Two constriction regions were identified by Su *et al* in the crystal structure of CmeC [32]: one at the periplasmic entrance, defined by residues Q412, D413, E416, and N420 in H9; and one at the extracellular exit, defined by residue R104 in L1. The channel diameter across the final 100 ns of the three equilibrium replicates was characterised using the analysis program HOLE [63]. Our findings were consistent with these constrictions, with the narrowest points in the channel aligning with residues E416 and R104 (**Fig 6A and 6B**).

The extracellular constriction displays considerable variation (**Fig 6A**). During the equilibrium simulations the L1 loops sample conformations both more and less 'open' than those in the crystal structure (**Fig 6C and 6D**). This is determined *via* the dihedral angles defined between the three N112 Cα atoms and a single R104 Cα atom (**Fig 6C**); in the crystal structure the average dihedral across the three protomers is 90.7˚, but over the course of the equilibrium simulations angles as small as 32.3˚ and as large as 130.1˚ are sampled. This is similarly observed in the simulation of TolC and OprM, though the loops of these homologues open to a greater extent and sample a greater range of angles [41,45]. This may be a result of the use of LOS in our system, where the bulky LOS headgroups and core oligosaccharides sterically hinder the further opening of the loops. Additionally, this reduced range may be due to the

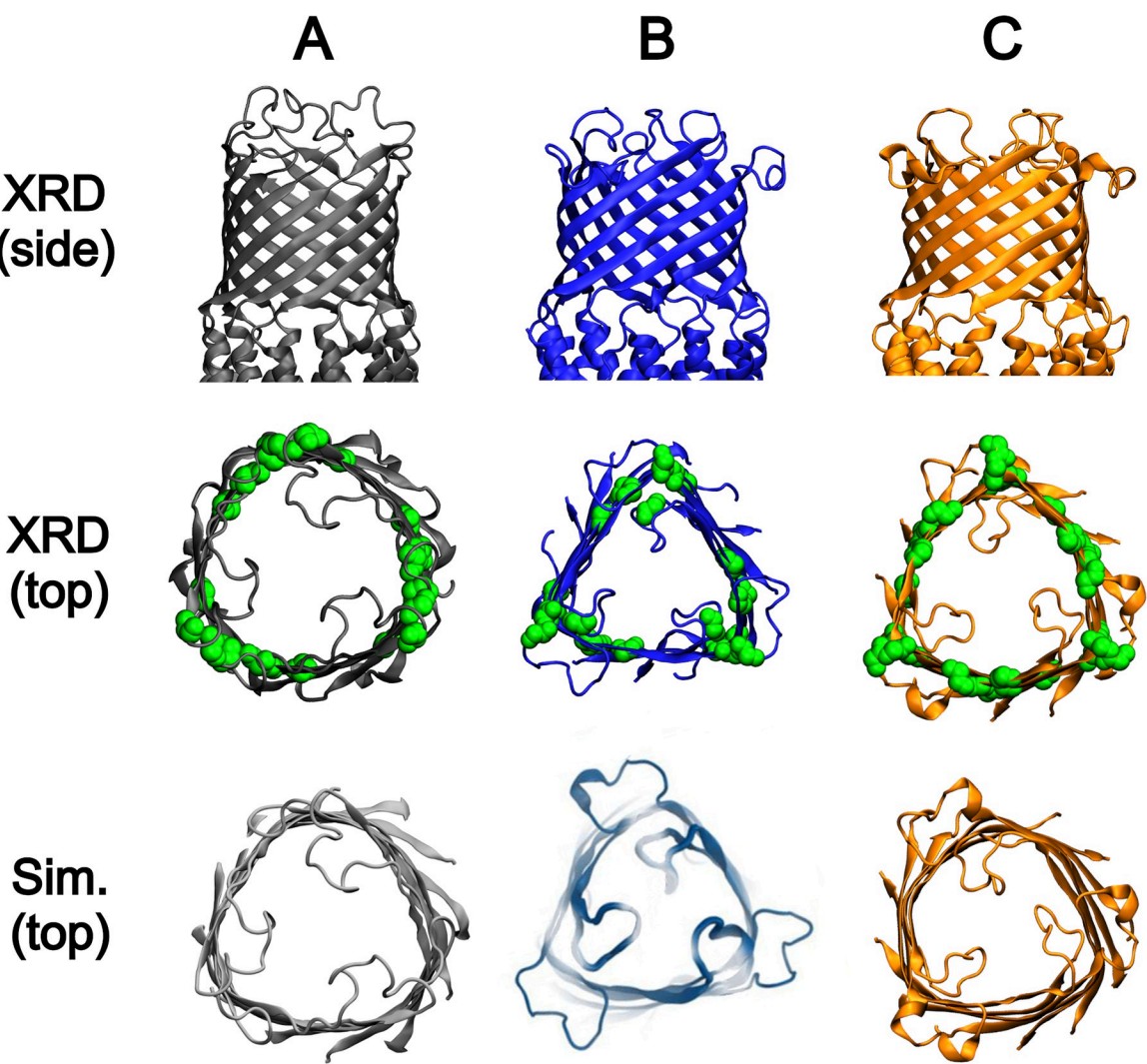

**Fig 5.** Shape of the β-barrels of the outer membrane factors **(A)** TolC, (PDB ID: 1EK9) **(B)** OprM, (PDB ID: 3D5K), and **(C)** CmeC (PDB ID: 4MT4). Top row: Side-view of the β-barrels of these proteins from their x-ray crystal structures. Middle row: Top-down view of the β-barrels of these proteins from their x-ray crystal structures. The positions of glycine residues in this domain are shown as green spheres. Bottom row: Top-down view of the β-barrels of these proteins after simulation in a model membrane. TolC adopts a distinctly triangular structure when simulated (from its cylindrical x-ray structure). OprM maintains its triangular prismatic structure across its simulation (Reprinted (adapted) with permission from Ref 45. Copyright 2013 American Chemical Society). CmeC starts as a triangular prismatic structure and appears to adopt an approximately hexagonal prismatic structure after simulation in a membrane. The 'corners' of each of the β-barrels aligns with the positioning of the glycine residues.

presence of the *N*-glycans, which are known to reduce protein dynamics [58]. Despite this, our data further support the hypothesis that OMFs are not gated at the extracellular exit as the loops freely explore open and closed conformations.

From the crystal structure, Su *et al* suggest a circular hydrogen bonding pattern in which residues E416 and Q412 of an adjacent protomer interact to form the core of the periplasmic constriction [32]. However, the salt bridge between the side chains of these residues is not persistent across the simulations; hydrogen bonding between the amide of Q412 and the carboxylate of a neighbouring E416 is observed in just 1% of the equilibrium trajectory frames. Despite this, the E416 and Q412 residues do still constitute the narrowest constriction of the channel with an average effective diameter of just 4–5 Å in this region. As shown in **Fig 6A**, there is

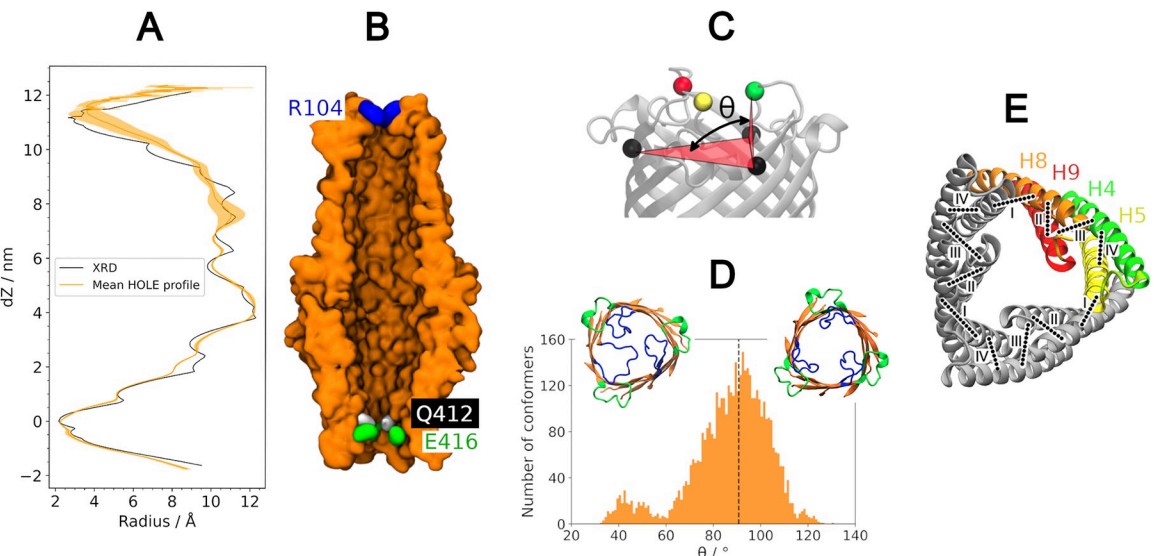

**Fig 6. Channel diameter constriction regions.** **(A)** HOLE profile of the CmeC channel. HOLE profile from the x-ray crystal structure shown in black, average profile across the final 100 ns of the three equilibrium simulations, with associated standard deviation, in orange. There is little variation is observed at the periplasmic constriction defined by residues E416 and Q412. The extracytoplsamic constriction, defined by the R104 residues of L1, shows more variation; the extracellular loops visit both the open and closed conformations freely. **(B)** cutaway of CmeC for scale for the HOLE profile. The two constriction regions align with the residues identified as constricting by Su et al. (30) **(C)** The dihedral angle determining the degree of opening of the L1 loops is defined by the three N112 alpha carbons (black) and one R104 alpha carbon (red, green, yellow). **(D)** Histogram of the dihedral angles observed over the three equilibrium simulations, with snapshots of loops more closed (left) and more open (right) than the XRD structure (black dashed line). **(E)** Circular inter- and intraprotomer hydrogen bonding network between the helices of the coiled coil domain. I: Interprotomer (H5-H8') salt bridges R370-E226, N385- E212, R395-E201. II: Intraprotomer (H8-H9) hydrogen bond W421- E390. III: Intraprotomer (H9-H4) Hydrogen bonding and salt bridges K432-E171, N429- N172, Q161-D441, N172/N168-N430. IV: Intraprotomer (H4-H5) salt bridge K222-E173.

very little variation in the channel radius in the periplasmic domain (coiled-coil and alpha bar-rel regions). Several intra- and interprotomer salt bridges and hydrogen bonds in a circular network may be promoting this closed conformation (**Fig 6E**). Intraprotomer hydrogen bonds were observed across the simulations between H9 and H4 (K432-E171, N429-N172, Q161-D441); H5 and H4 (K222-E173); H9 and H8 (W421-E390). Interprotomer salt bridges R370-E226', R395-E201', and N385-E212' (H8-H5') were also observed in every replicate.

Previous simulation and mutational studies of OMFs have suggested an 'iris-like' motion of the coiled-coil domain to allow the periplasmic entrance to open, requiring a major conforma-tional rearrangement at a high energetic cost. Mutational and Markov state modelling studies have demonstrated that for the periplasmic entrance of TolC to open, interprotomer interac-tions must be disrupted, but intraprotomer interactions may remain intact [39,44]. We do not observe a twisting motion in any of the three equilibrium replicates, nor do we observe any correlated disruption of the inter- and intraprotomer hydrogen bonding network.

Despite simulating this system for a total of 1.5 µs, our simulations are still relatively short for such a large and complex system: simulation studies of TolC in smaller and simpler mem-brane systems demonstrated that submicrosecond simulations were insufficient to adequately sample the equilibrium behaviour of the periplasmic gate [43]. To prevent the uncontrolled diffusion of solutes, OMFs adopt a closed conformation when not coupled to their periplasmic adaptor partners: *in vivo* this protein is coupled to the PAP CmeA, and likely interacts with the peptidoglycan cell wall [53,64,65] and potentially other proteins/efflux machinery [66]. We have simulated CmeC as an isolated membrane protein so it is no surprise that we only observe

the closed state under equilibrium conditions. Another factor is the cations in our model, which is discussed in the following section.

## Cation binding sites within the channel

Cation binding sites have been identified in the OMFs TolC and OprM [67,68], and simulations have shown that for the periplasmic end of the wild-type proteins to open *in silico* these cations must be removed from the system [41,45]. Cations have also been identified as potential orthosteric inhibitors of outer membrane factors due to their strong coordination by the conserved rings of acidic residues in the coiled-coil domain [67,69,70]. Here we identify regions within CmeC that display substantial cation density across the equilibrium simulations, and observe cations contributing to the closed conformation of the coiled coil domain.

Initially we expected the solvated potassium ions to be the prime candidate for cation coordination within the channel. However, we identified very little potassium ion density within CmeC (**S8 Fig**). Unlike previous simulations studying OMF dynamics (which have used symmetric phospholipid bilayers and $Na^+$ and $Cl^-$ ions in solution) our model included LOS in the outer leaflet with divalent calcium ions (necessary for outer membrane integrity [71–81]) in addition to $K^+$ and $Cl^-$ in solution; using the VMD plugin VolMap we identified $Ca^{2+}$ density within the CmeC channel across the equilibrium simulations (**Fig 7A**). During equilibration and production stages of the simulations calcium ions have escaped the LOS headgroups, and due to periodic replication of the simulation system were able to enter the protein from both the extracellular and periplasmic ends. Due to the increased coulombic attraction between $Ca^{2+}$ and the carboxylate residues (compared to $K^+$), calcium ions are coordinated preferentially. The calcium ion density is primarily situated proximal to acidic residues D413, E416, E353, E107, D86, and D77. **Table 1** lists the proportion of equilibrium trajectory frames in which a calcium ion was coordinated by each of these acidic residues (in at least one protomer).

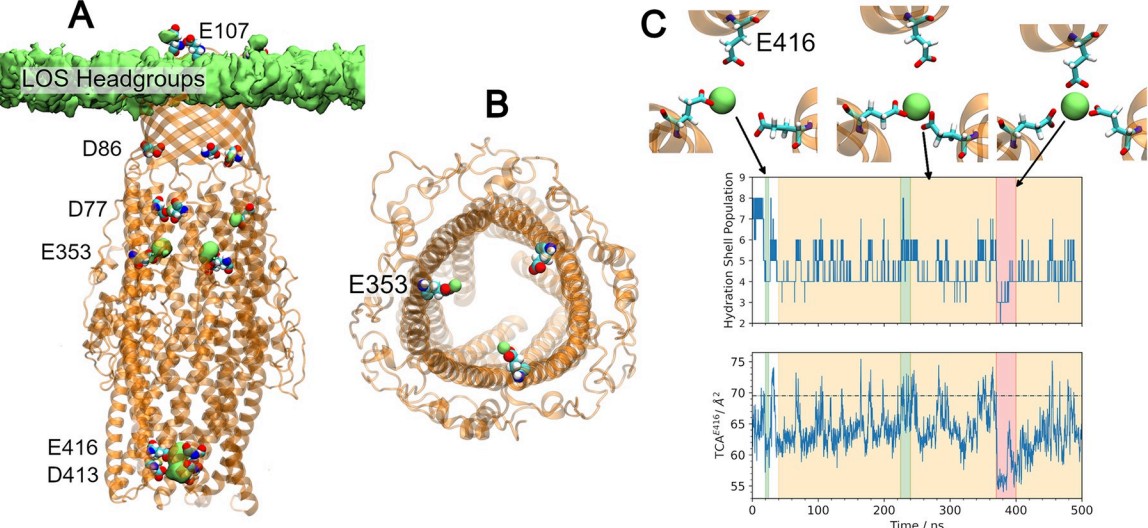

**Fig 7. Acidic residues within the channel coordinate calcium ions. (A)** Calcium ion density (green surface) located close to acidic residues (VDW, labelled) within the CmeC channel. Considerable calcium ion density was found near residues D413, E416, E353, D77, D86, and E107. **(B)** Acidic residues above the coiled-coil region coordinate calcium ions in a 1:1 stoichiometry. Here, two calcium ions (green spheres) are bound to two of the three E353 residues. **(C)** The calcium ions bound to E416 see drastic changes to their hydration shell as 1 (green), 2 (yellow) or all 3 (red) acidic residues coordinate a single cation. Coordination is coupled with the triangular cross-sectional area: as more E416 residues coordinate, the triangular cross-sectional area defined by the E416 alpha carbons decreases. TCA defined in the x-ray crystal structure shown as a dashed black line.

**Table 1. Proportion of trajectory frames in which selected acidic residues are observed to coordinate a calcium ion.**

| Residue | % Frames containing interaction | | Change in % |
|---|---|---|---|
| | Equilibrium MD | CDCA steered MD | |
| D413 | 62 | 72 | +10 |
| E416 | 96 | 99 | +3 |
| E220 | 16 | 25 | +9 |
| E107 | 91 | 97 | +6 |
| E353 | 89 | 92 | +3 |
| D86 | 24 | 42 | +18 |
| D77 | 32 | 91 | +59 |

Calcium ions may be coordinated by up to three acidic residues simultaneously (one from each protomer). For aspartate/glutamate residues in the β- and α-barrel regions of the protein (e.g. E353), the internal diameter of CmeC is such that the acidic residue from only one protomer can coordinate a single calcium ion at any given time (**Fig 7B**). By contrast, the lower coiled-coil region is sufficiently narrow that acidic residues from multiple protomers can coordinate the same calcium ion; in all replicates, two E416 residues consistently coordinate a single calcium ion, and occasionally all three residues will coordinate this central ion (**Fig 7C**). This additional level of coordination results in calcium ions remaining bound in this region for hundreds of nanoseconds at a time.

We note here that previous computational studies have identified systematic overestimation of protein-ion affinities for cations in the CHARMM force fields using single-point cation models [82,83]. Updates to the force field in late 2020 attempted to rectify such issues by increasing the Lennard-Jones parameter, $\sigma$, for carboxylate-cation interactions [84]. A similar change to non-bonded parameters has been shown to improve binding affinities dominated by cation-$\pi$ interactions [85]. We have used the CHARMM36m force field with these updated Lennard-Jones parameters; to our knowledge, this model is appropriate for the scope of this paper. However, should binding free energies be calculated, a polarisable or multi-point ion model would be most appropriate.

Coordination of calcium ions by acidic residues near the periplasmic entrance causes a contraction in the triangular cross-sectional area defined by C$\alpha$ of the E416 residues (TCA$^{E416}$) (**Fig 7C**). The increased coordination is also reflected in the reduced solvation sphere (number of water molecules with at least one constituent atom within 3 Å of the ion's centre of mass) for the bound calcium ion. A calcium ion in the CHARMM36m force field using the TIP3P water model displays an average solvation sphere of 7–8 water molecules, in agreement with experimental data [86]. Coordination to one, two, or three acidic residues within CmeC results in solvation spheres of 5–6, 4–6 and 2–3 water molecules respectively (**Fig 7C**).

To further assess the link between this coordination and tight periplasmic constriction, additional simulations were undertaken in which the cations within the channel were removed. Chloride ions were also removed from solution to neutralise the net positive charge, and these new systems were simulated in duplicate. When only calcium ions were removed (and remaining calcium ions restrained to prevent re-entry into the channel), there was a small increase in the average TCA$^{E416}$ (**Table 2**). In these simulations, potassium ions can

**Table 2. Average TCA$^{E416}$ (with associated standard deviation) from the simulations.** For the equilibrium simulations, TCA is calculated from the final 100 ns over the three replicates. For systems where cations have been removed, TCA is calculated from the final 20 ns over the two replicates.

| | Equilibrium simulations | Ca$^{2+}$ removed | Ca$^{2+}$ and K$^+$ removed |
|---|---|---|---|
| TCA$^{E416}$ / Å$^2$ | 63.40 ± 3.82 | 67.09 ± 4.23 | 115.36 ± 9.49 |

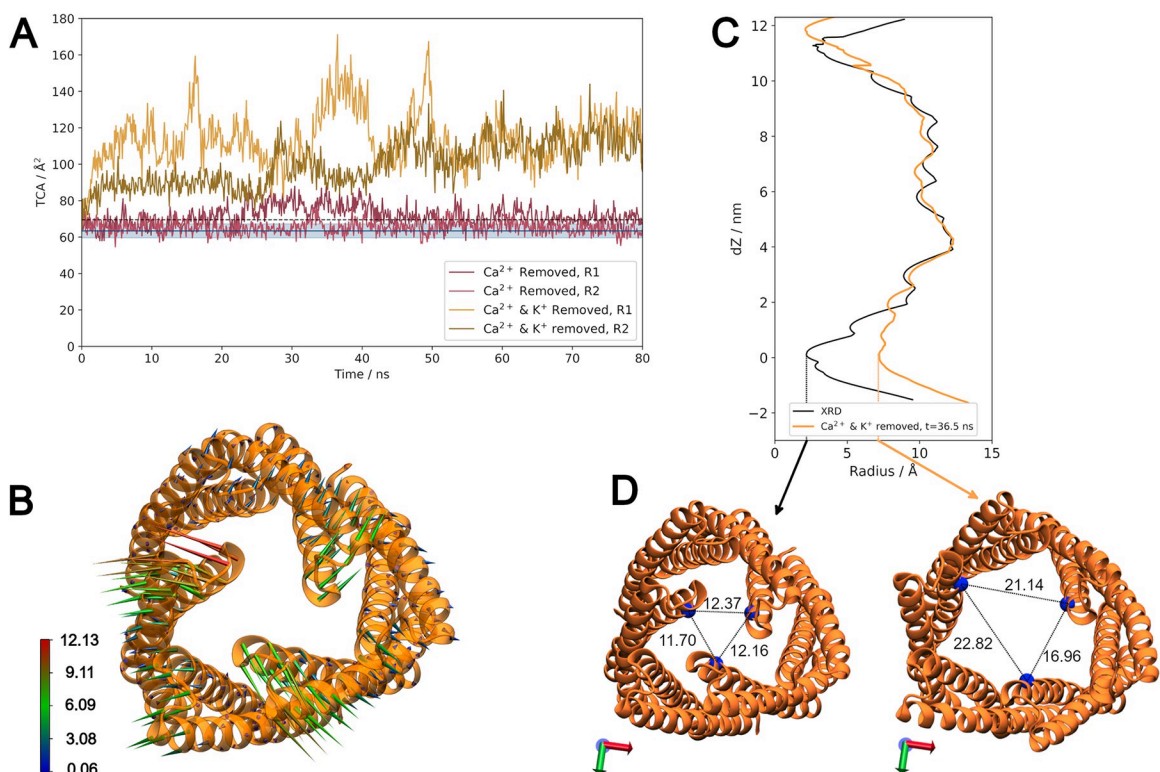

**Fig 8. Cation removal allows the coiled-coil domain to open. (A)** Timeseries of TCA$^{E416}$ when only calcium ions are removed from the channel and remaining calcium ions restrained (pink), and when all cations within the channel are removed and all remaining cations restrained (yellow). Removing all cations within the channel allows the coiled coil region to open. **(B)** Porcupine plot indicating the direction and magnitude of motion between the initial, closed conformation of CmeC to the most open structure observed in these simulations (Ca$^{2+}$ & K$^+$ removed, replicate 1 at t = 36.5 ns). Bottom-up view of the coiled-coil domain. Colour scale from 0.06 Å (blue) to 12.13 Å (red). **(C)** Channel radius profile for the most open conformation, with the XRD profile for comparison. The channel radius at the periplasmic entrance has increased considerably. **(D)** Bottom-up view of the periplasmic entrance to CmeC in the initial equilibrated conformation (left) and the most open conformation (right). Distances between E416 Cα atoms of E416 residues displayed (Å).

become coordinated at sites previously occupied by calcium ions. This includes the rings of acidic residues at the periplasmic constriction; in lieu of calcium ions, potassium ions can hold this end of the protein in a closed conformation.

When both potassium and calcium ions were removed (and all remaining cations restrained) there was a large increase (+82%) in the average TCA$^{E416}$ (**Fig 8A**) as there is no longer a central cation holding the coiled-coil tightly closed. This is also reflected in the protein backbone RMSD (**S9 Fig**) and the pore radius profile (**Fig 8C**). This increase is consistent with simulation studies in which the removal of sodium ions at the periplasmic end of TolC and OprM caused an increase in TCA at D374 and D416 (both corresponding to E416 in CmeC) respectively [41,45]. This increase in TCA is driven primarily by the outwards motions of H8 and H9, as shown in **Fig 8B**. For further opening of the coiled coil domain in an iris-like fashion, interprotomer interactions must be disrupted [43,44]; as with our equilibrium simulations we do not observe any correlated disruption of interprotomer (or intraprotomer) hydrogen bonding networks in the coiled-coil domain (**S10 Fig**). These data further reinforce the hypothesis that the periplasmic entrance of OMFs is regulated both by interprotomer interactions [43,44] and by cations [37,45].

## Substrate translocation

CmeABC is known to extrude a variety of structurally unrelated compounds, including various antibiotics and bile salts [27,87–90]. To investigate the translocation of substrates through CmeC we used steered molecular dynamics (SMD) to pull erythromycin (a macrolide antibiotic and the therapy of choice for *Campylobacter* infections [91]) and chenodeoxycholic acid (CDCA, a bile acid commonly found in avian species [92]) through the channel (**Fig 1C and 1D**).

Both substrates successfully entered and translocated the channel with the aid of the external pulling force. Force profiles for both substrates indicate that entry to CmeC is the greatest barrier to translocation (**Fig 9A**). The force required to pull CDCA is greater than that for erythromycin by around 50%, despite erythromycin being a larger molecule. In aqueous solution erythromycin is uncharged, whereas CDCA is deprotonated to yield an anionic species. This constriction in CmeC is primarily acidic residues (most notably E416 and D413), and the interior surface of CmeC is 'strikingly electronegative' [32] (**Fig 1B**); CDCA faces a greater barrier to entry in the form of electrostatic repulsion.

It is of note however that we did not observe either substrate to enter the channel *via* the conventional route: we would expect the substrates to enter the channel through the centre, but instead we observed these molecules to enter in gaps defined by the periplasmic ends of helices H4, H5, H8 and H9 (**Fig 10A**). This resulted in a temporary increase in TCA$^{E416}$ as helices moved to accommodate the substrate (**Fig 10B**). Despite this, the translocation of the substrates did not appear to disrupt the intra- and interprotomer hydrogen bonding networks, though E416-Ca$^{2+}$ coordination was disrupted. As the substrates move through this constriction, H9 moves outwards and the glutamate residues from different protomers are separated; only one E416 coordinated a calcium ion during this time, if at all.

When simulated in the CHARMM36m force field using the TIP3P water model, erythromycin has an average solvation sphere of ~57 water molecules, and that of CDCA is ~39 waters. In order to enter and translocate the channel the substrates must shed water molecules. The decrease in water molecules around each substrate is heavily correlated to the pore radius (**S11 Fig**); at the narrowest constriction point, the average number of water molecules around erythromycin decreases by 35%, and around CDCA by 49% (**Fig 9B**). As the channel increases in radius, hydration increases for both substrates, though the solvation sphere contains fewer waters than in solution at all points during translocation.

## Substrate orientation preference

It has previously been shown that outer membrane proteins can enforce orientation requirements for substrate passage [49,93]. By overlaying every n$^{th}$ frame from the steered MD trajectories we can visualise how the orientation of each substrate changes during translocation. Erythromycin shows no clear preferences/requirements in any of the SMD simulations: there are no regions within CmeC in which erythromycin displays a consistent orientation (**Fig 10C**). An additional plot was generated to show the clustering of a particular atom (a hydroxyl oxygen) over all of the trajectories (**S12 Fig**). Erythromycin is polar, but uncharged at physiological pH. Its polar functional groups (hydroxyls, esters, and an amine) are distributed across the molecule, which is reflected in the electrostatic profile of this molecule (**S13 Fig**) As such, erythromycin is capable of hydrogen bonding with the channel residues regardless of its orientation, thus showing no clear preference on translocation.

CDCA has only three polar groups: two hydroxyl groups, and a carboxylic acid moiety which is deprotonated at physiological pH. The resulting anionic carboxylate moiety (reflected in the electrostatic profile, **S14 Fig**) will therefore dominate the interactions between CDCA

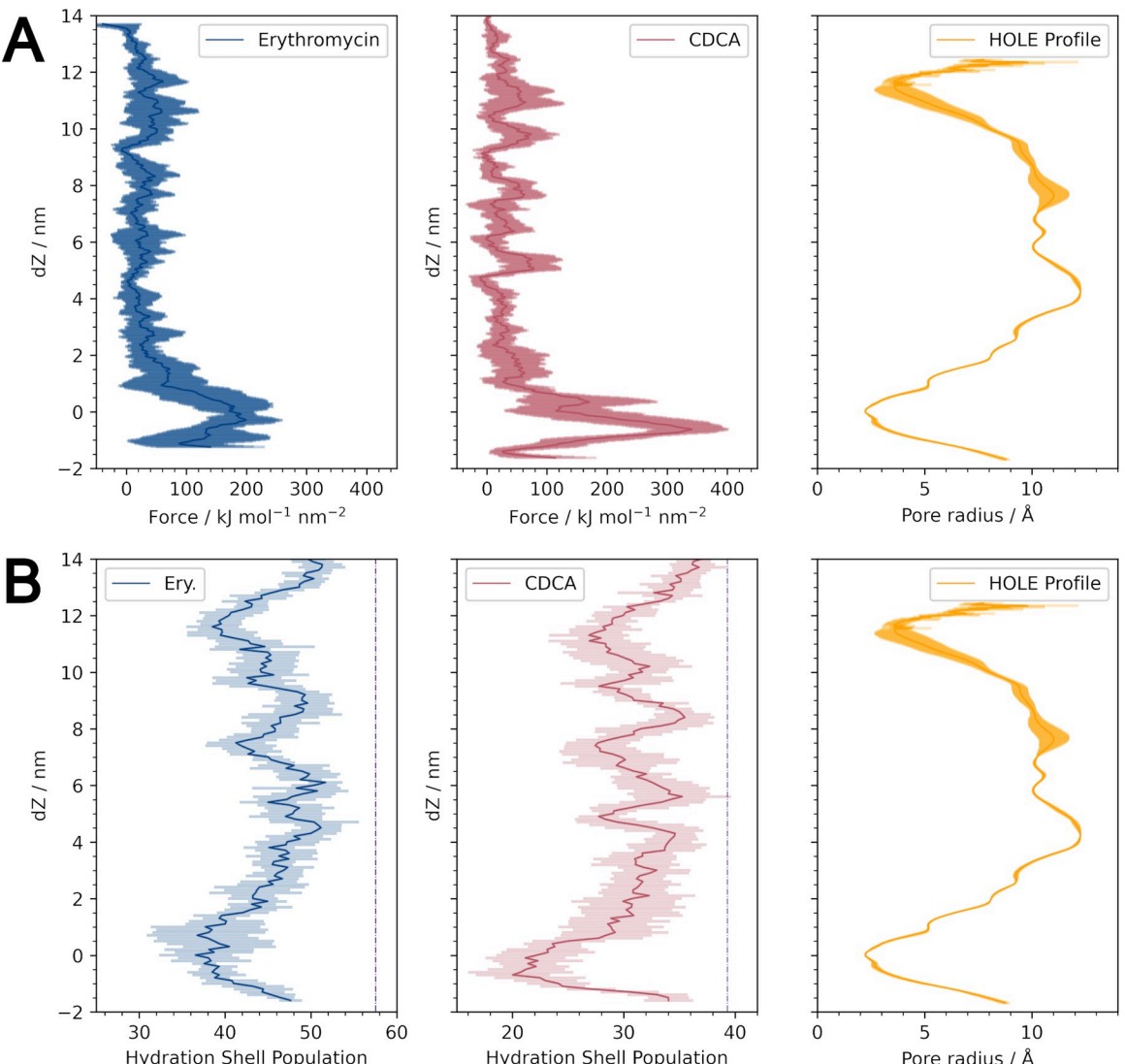

**Fig 9. Force and substrate hydration profiles for translocation of erythromycin and CDCA through CmeC. (A)** Average force profile for the translocation of each substrate with associated standard deviation, with the HOLE profile for reference. The greatest barrier to translocation is the tight periplasmic entrance. There is no great force barrier at the extracellular exit, implying that there is no gating at this end. **(B)** Average hydration profiles for the substrates as they translocate the channel with associated standard deviation, with the HOLE profile for reference. Average hydration shell population in solution indicated for each substrate as a vertical dashed line. Both substrates must lose a considerable proportion of their hydration shell to translocate, with the greatest decrease at the periplasmic entrance.

and the predominantly anionic interior surface of CmeC. Across the nine SMD simulations in which this substrate is pulled through the channel, CDCA shows orientation preference at several positions; the carboxylate moiety of the molecule orients towards the side chains of channel-lining residues (**Fig 11A**). An additional plot was generated to show the clustering of a particular atom (a carboxylate oxygen) over all of the trajectories (**S15 Fig**). Interestingly, these regions of preferred orientation correlate with the positions of acidic residues that line the channel. This is explained by the presence of the multivalent $Ca^{2+}$ ions in the system: the carboxylate moieties of both CDCA and aspartate/glutamate residues can simultaneously coordinate a single calcium ion (**Fig 11D**). Across the 9 trajectories, we observe CDCA to coordinate a calcium ion in 64% of frames, with an average coordination lifetime of 3.6 ns (maximum

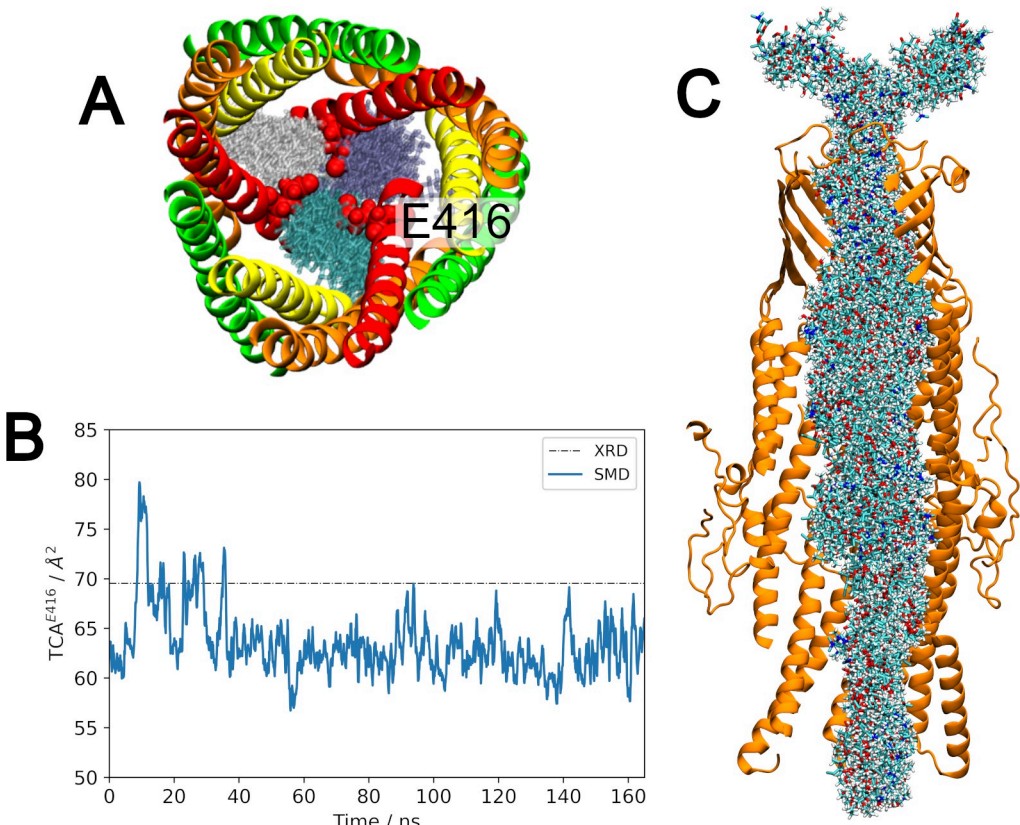

**Fig 10. Translocation of Erythromycin through CmeC. (A)** Entry of erythromycin (shown as overlaid white, purple, and cyan transparent structures) occurs between the coiled-coil helices (shown in red, orange, green, and yellow New Cartoon representation) rather than through the central constriction defined by the side chains of the E416 residues (shown in red VDW). This entry pathway is also observed for CDCA. **(B)** Timeseries of the triangular cross-sectional area defined by the Cα atoms of the three E416 residues (TCA$^{E416}$). Erythromycin enters the channel within the first 35 ns, which disrupts the coordination of E416 residues to the central calcium ion, causing the protomers to move apart and the TCA to increase. When erythromycin has passed this constriction region the TCA returns to a value below that in the x-ray structure as multiple E416 residues coordinate the central calcium ion. **(C)** Cutaway of CmeC (orange New Cartoon representation) with every 5th frame for erythromycin (licorice, coloured by atom name) from three replicates overlaid. Hydrogen atoms not shown for clarity. Erythromycin displays no clear orientation preference on translocation.

observed lifetime of 23.5 ns). 21% of these coordination interactions were transient (lasting less than 0.5 ns), though these short interactions constitute just 1% of the total interaction time. In turn we observe these CDCA-coordinated calcium ions to be simultaneously coordinated by acidic residues lining CmeC in 20% of the total frames. When these CDCA-Ca$^{2+}$-ASP/GLU interactions are broken on CDCA translocation, CDCA abstracts the calcium from the acidic lining residue in 67.7% of unbinding events.

The presence of CDCA within the channel also appears to promote the coordination of calcium ions to these acidic lining residues: as shown in **Table 1** the proportion of frames in which a calcium ion is coordinated is somewhat increased in the SMD trajectories when compared to the equilibrium simulations. The phenomenon of cation coordination increasing to cationic to acidic residues in the presence of CDCA within the channel is striking and reproducible. While these changes are statistically significant for D77 and E416 (p values of 0.00171 and 0.00672 respectively) longer trajectories/larger data sets are needed to confirm whether the increase in ion coordination is a general phenomenon or specific to some sites. We note

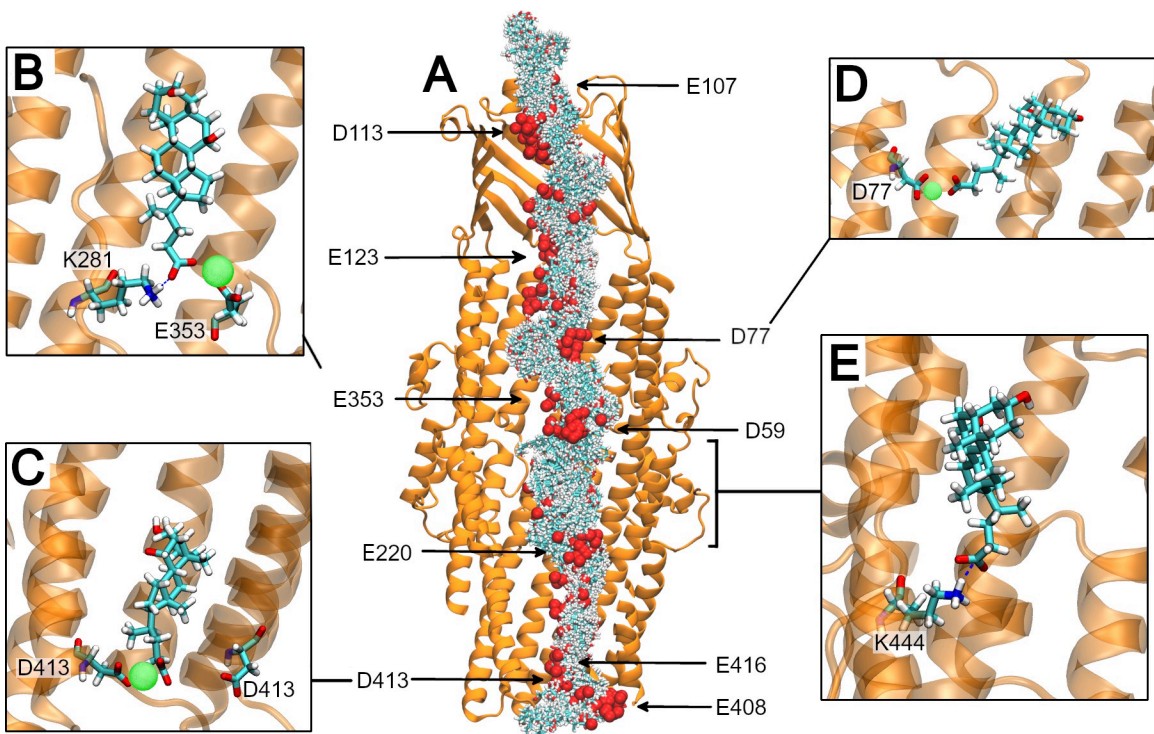

**Fig 11. Translocation of CDCA through CmeC. (A)** Cutaway of CmeC (orange New Cartoon representation) with every 5[th] frame for CDCA (licorice, coloured by atom name, with carboxylate oxygen atoms shown as spheres) overlaid from one representative SMD simulation. CDCA displays orientation preference in several regions, generally when close to acidic lining residues. **(B)** A snapshot from a steered MD simulation. CDCA can coordinate calcium ions that are bound to acidic residues within the CmeC channel. They can simultaneously hydrogen bond to nearby basic residues, those these interactions are most commonly transient. **(C)** A snapshot from an equilibrium simulation. CDCA coordinates a calcium ion bound at the periplasmic entrance. This interaction with the tightly bound calcium is much longer-lived than interaction with bound calcium ions at wider points in the channel. **(D)** A snapshot from a steered MD simulation. CDCA coordinates an aspartate-bound calcium as it translocates. **(E)** A snapshot from an equilibrium simulation. In this replicate CDCA did not coordinate a calcium ion, and instead formed a salt bridge with K444 multiple times, with an average lifetime of 17 ns.

that E416 is part of a ring of acidic residues at the periplasmic entrance that is conserved across OMFs, while D77 is not conserved.

We do observe salt bridges between CDCA and the basic residues lining CmeC, though only in 6.3% of trajectory frames. Basic residues are found predominantly in the α- and β-barrel domains and often positioned close to an acidic residue where their side chains become less readily available to translocating substrates by engaging in salt bridges. Additionally, by the time CDCA reaches these basic residues it is often already coordinating a calcium ion, making interaction with these cationic side chains less favourable.

Full umbrella sampling with adequate convergence for the calculation of a PMF profile was deemed unfeasible for such a long and wide channel. Instead, several equilibrium simulations were initiated from snapshots with CDCA within the channel to further assess CDCA-CmeC interactions. A separation of approximately 1 nm was used between windows, with additional replicates where CDCA is proximal to the acidic residues where an orientation preference was observed in SMD simulations. In these equilibrium simulations we observe a similar trend in interactions with the acidic lining residues. Over the course of the 23 equilibrium simulations (totalling 2.78 μs) CDCA coordinates a calcium ion in 78% of frames, with 39% of the total frames displaying a calcium ion coordinated to both CDCA and an acidic residue simultaneously.

The mean lifetime of CDCA-$Ca^{2+}$ coordination is 28 ns. 13% of interactions were transient (but constituting only 0.1% of the total interaction time), and coordination lifetimes of up to 151 ns were observed. In turn, these CDCA-coordinated $Ca^{2+}$ ions are also coordinated by ASP/GLU residues. On average, CDCA remains coordinated to the GLU/ASP-bound calcium ions for 7 ns. 31% of these interactions were transient, constituting 0.8% of the total interaction time. A maximum interaction lifetime of 106 ns was observed where CDCA coordinated a calcium bound to the E416 and D413 residues at the periplasmic entrance (**Fig 11C**); the four replicates that included this interaction saw an average lifetime of interaction of 22 ns, with CDCA coordinating this tightly bound calcium in 80% of frames. When CDCA-$Ca^{2+}$-GLU/ASP interactions are disrupted the calcium ion can remain bound to the GLU/ASP residue, or be abstracted by CDCA. In 61.4% of unbinding events CDCA abstracts the calcium.

Interaction with basic residues is observed, though again these interactions are less prevalent than those with the acidic residues (6.2% of frames). In the majority of cases the interaction with basic residues is transient and/or occurs simultaneously with coordination to a calcium and nearby acidic residue (**Fig 11B**). The exception to this is a single equilibrium simulation in which the starting conformation places CDCA (without a coordinated calcium ion) proximal to K444 (**Fig 11E**). In this case CDCA is initially free in solution within the channel, interacts with K444 multiple times across the simulation (average lifetime of 17 ns), and is not observed to coordinate a calcium ion (and in turn interact with acidic residues) despite being proximal to D441. However, orientational preference around this basic residue was not observed in the SMD simulations; the significance of this interaction remains to be determined.

## An acidic ladder for an anionic substrate?

Taking these results in combination, we can begin to build a picture for how the anionic CDCA translocates this predominantly anionic channel. Equilibrium MD showed that acidic sites within the channel can coordinate calcium ions from solution, partially neutralising the anionic nature of the channel lining. When CDCA is steered through the channel, we observe its carboxylate moiety to coordinate these bound calcium ions, as well as calcium ions in solution, resulting in several regions of preferred orientation for the CDCA molecule within the channel. CDCA often abstracts this calcium ion on translocation (67.7% of unbinding events in SMD, 61.4% in equilibrium MD). We propose that CDCA is utilising the calcium ions to climb the ladder of acidic residues that line the interior surface of CmeC.

There is precedence for such a ladder mechanism in the translocation of arginine through the outer membrane protein OprD from *P. aeruginosa* [93]: steered MD, docking, and equilibrium molecular dynamics revealed a distinct orientational requirement along the permeation pathway in which the carboxylate moiety of free arginine interacted with a previously-identified basic ladder of arginine residues. Here we note that while CDCA does not display an orientational *requirement* for translocation, there are clear areas of preference observed in all the SMD replicates that suggest this coordination to protein-bound calcium ions is important. Whether this mechanism of translocation applies to other anionic substrates, such as the fluoroquinolone family of antibiotics, and how cationic substrates traverse the channel remain to be investigated.

## Limitations

As with any biomolecular simulation study, adequate sampling is of concern. Even after 500 ns of simulation, CmeC has not reached an equilibrium conformation in any of the three equilibrium replicates. Longer simulations are required to reach an equilibrium state, but given the

size of this system it is unclear how long this may take: even when a simulation was extended to 1.25 µs the RMSD was still rising. However, principal component analysis indicates that the equilibrium simulations sampled different conformational states, unique from the X-ray structure.

We have simulated CmeC as an isolated membrane protein, when *in vivo* this protein would be coupled to CmeB *via* CmeA and likely interacts with the peptidoglycan cell wall. OMFs are known to adopt a closed state when not coupled to their PAP partner to avoid uncontrolled movement of species into and out of the cell; in the presence of cations we have only observed the closed state of CmeC during our simulations. CmeC may also interact with other OMPs or efflux systems [66] which have been omitted. Our outer membrane model, despite modelling the LOS in the outer leaflet, is likely still a simplification: in 2020 Cao *et al* demonstrated that the phospholipodome of *C. jejuni* contained a substantial proportion of lysophospholipids [56], which may influence protein structure and dynamics but have not been included in our model.

Whenever considering drug/small molecule binding to a protein or translocation pathways it is desirable to compute free energies for unambiguous comparisons. We have performed a number of steered MD simulations here and the temptation is to exploit the Jarzynski equality to extract free energies from the force profiles [94]. However the large system size (close to 1 million atoms) [95], insufficient simulation lengths with only a single pulling rate [96,97], and the importance of electrostatics in the protein-substrate interactions [96] precludes this.

## Conclusions

The equilibrium simulations presented here show that CmeC behaves in a similar fashion to other simulated OMFs TolC and OprM: we observe a periplasmic gate which displays very little dynamic nature, held tightly closed by cations and inter- and intraprotomer hydrogen bonding networks; and a highly dynamic extracellular constriction in which extracellular loops visit both open and closed conformations. The *N*-glycans display greater flexibility than previous studies have suggested. Within the channel, we identify several acidic residues that coordinate calcium ions from solution and show that the coordination of cations in the coiled coil region contributes to the closed conformation of CmeC. Steered MD showed that the acidic sites lining the channel were areas in which the anionic substrate CDCA displayed a clear orientation preference. CDCA was observed to coordinate the bound calcium ions, and we propose that this substrate coordinates these calcium ions to 'climb' its way up the channel interior.

## Methods

All molecular dynamics simulations were performed in the CHARMM36m force field [98,99] with TIP3P water [100] in the GROMACS 2019.4 package [101,102]. The LINCS algorithm was utilised to constrain all bonds [103]. Van der Waals interactions were smoothed at distances beyond 1.0 nm to a cut-off at 1.2 nm. Long-range electrostatics were treated using Particle Mesh Ewald [104] with a cut-off of 1.2 nm. The system was coupled to a heat bath at 315.15 K using the velocity-rescale thermostat [105] at equilibration and production stages ($\tau_T$ = 1.0 ps). Analyses were performed using GROMACS and MDAnalysis utilities [106–109]. Molecular graphics were generated in VMD 1.9.4a51 [110].

### System construction

The CmeC X-ray structure was obtained from the RCSB Protein Data Bank (accession code: 4MT4) [32]. Missing C-terminal residues were added using MODELLER [111]. The

**Table 3. Position restraints, timestep, and duration of the equilibration phases.**

| Equilibration Phase | Position restraints / kJ mol$^{-1}$ nm$^{-2}$ | | | | dt / fs | Duration / ns |
|---|---|---|---|---|---|---|
| | Protein Backbone | Protein Sidechain | Lipid Headgroups | Dihedrals | | |
| NVT1 | 4000 | 2000 | 1000 | 1000 | 1 | 0.125 |
| NVT2 | 2000 | 1000 | 400 | 400 | 1 | 0.125 |
| NPT1 | 1000 | 500 | 400 | 200 | 2 | 0.5 |
| NPT2 | 500 | 200 | 200 | 200 | 2 | 0.5 |
| NPT3 | 200 | 50 | 40 | 100 | 2 | 0.5 |
| NPT4 | 50 | - | - | - | 2 | 0.5 |

CHARMM-GUI Membrane Builder Module [55,112] was used to modify the protein structure with N-terminal lipidation and appropriate N-glycosylation at residues N13 and N30 in each protomer, then to embed the β-barrel domain of CmeC in a model *C. jejuni* outer membrane (1:1 ganglioside mimicry 1 and 1a LOS in the outer leaflet, and 4:1 POPE and POPG in the inner leaflet) [54,55]. This protein-membrane system was solvated in 150 mM KCl with additional calcium ions (equating to a concentration of 125 mM) associated with the LOS core sugar phosphates. The system was energy minimised in 50,000 steps using the steepest descent algorithm [113], then equilibrated sequentially using 2 NVT phases and 4 NPT phases with decreasing position restraints at each phase (**Table 3**). All NPT phases coupled the system to a pressure bath at 1 bar using the semi-isotropic Berendsen scheme [114] ($\tau_p$ = 5.0 ps, compressibility = 4.5x10$^{-5}$ bar$^{-1}$).

## Equilibrium molecular dynamics

Three replicates of this system were generated. One simulation used the final frame of the 6th equilibration phase as the initial conformation. The additional replicates were each heated to 330 K for 20 ns with position restraints of 4,000 and 2,000 kJ mol$^{-1}$ nm$^{-2}$ applied to the protein backbone and sidechains respectively to induce new starting configurations for the membrane lipids. Each system was simulated at 315.15 K for 500 ns. Pressure was maintained at 1 bar using the semi-isotropic Parrinello-Rahman coupling scheme [115,116]. The contents of this system, and all subsequent simulation systems, can be found in Table 4.

**Table 4. Number of each species present in each simulation system.**

| | Equilibrium MD | Ca$^{2+}$ removal | Ca$^{2+}$ & K$^+$ removal | Erythromycin SMD | CDCA SMD/MD |
|---|---|---|---|---|---|
| CmeC Protomers | 3 | 3 | 3 | 3 | 3 |
| LOS GM1 | 108 | 108 | 108 | 108 | 108 |
| LOS GM1a | 108 | 108 | 108 | 108 | 108 |
| POPE | 551 | 551 | 551 | 551 | 551 |
| POPG | 136 | 136 | 136 | 136 | 136 |
| K$^+$ | 841 | 841 | 827 | 841 | 842 |
| Cl$^-$ | 669 | 651 | 637 | 669 | 669 |
| Ca$^{2+}$ | 702 | 693 | 693 | 702 | 702 |
| TIP3P water | 245,701 | 244,701 | 244,701 | 244,701 | 244,700 |
| Erythromycin | 0 | 0 | 0 | 1 | 0 |
| CDCA | 0 | 0 | 0 | 0 | 1 |
| Total Atoms | 964,998 | 964,971 | 964,943 | 965,116 | 965,063 |

## Cation removal

The final frame of one replicate of the equilibrium system was extracted and used as the starting conformation for further simulations to assess the role of cations in the constriction at the periplasmic entrance. Two systems were generated: in the first, just calcium ions within the channel were removed (remaining calcium ions restrained, 1,000 kJ mol$^{-1}$ nm$^{-2}$), and in the second all calcium and potassium ions within the channel were removed (all remaining cations restrained, 1,000 kJ mol$^{-1}$ nm$^{-2}$). Chloride ions were removed from solution to neutralise the systems. Both systems were energy minimised in 5,000 steps using the steepest descent algorithm. Each system was simulated in duplicate for 80 ns.

## Steered molecular dynamics

Erythromycin and chenodeoxycholic acid (CDCA) molecules were generated in the CHARMM36m force field using CHARMM-GUI's Ligand Modeler module [117] and energy minimised in 5,000 steps using the steepest descent algorithm. Each molecule was manually positioned below the periplasmic entrance to CmeC in three different orientations. These protein-substrate structures were subsequently modified with the appropriate N-glycans and N-terminal lipids, embedded in a model membrane and solvated as described in the previous system construction. The systems were energy minimised and equilibrated using the same protocol as the equilibrium systems.

A harmonic spring of force constant 1,000 kJ mol$^{-1}$ nm$^{-2}$ was attached to the centre of mass of each substrate. This was pulled along the z-axis (perpendicular to the membrane) at a constant velocity of 0.1 nm ns$^{-1}$. No restraints were applied to the substrates in the x and y dimensions. A total of nine replicates were generated for each system, with each simulation lasting 160–170 ns. For the CDCA system, snapshots were taken with CDCA at various positions within the channel, and these were used as initial configurations for subsequent equilibrium simulations, each run for a minimum of 100 ns; a total of 23 equilibrium simulations were run with CDCA within the channel.

## Electrostatic profiles of substrates

The electrostatic potentials of erythromycin and the deprotonated CDCA molecules were calculated using the density functional theory package, ONETEP [118,119], using the PBE exchange-correlation functional [120], augmented with Grimme's D2 dispersion correction [121]. Open boundary conditions *via* real-space solution of the electrostatics were used in a simulation cell with dimensions 5 nm × 5 nm × 5 nm. Norm-conserving pseudopotentials were used for the core electrons, and the psinc basis set, equivalent to a plane wave basis set with a kinetic energy cut-off of 800 eV, was employed. 8.0 Bohr localisation radii were used for the nonorthogonal generalised Wannier functions (NGWFs).

## Supporting information

**S1 Fig. Chemical structure of the heptasaccharide attached to N13 and N30 in CmeC.** (TIF)

**S2 Fig. Timeseries data of the simulation box dimensions.** X and Y dimensions of the box are coupled (semi-isotropic pressure coupling used). Box dimensions are stable after ~300 ns. (TIF)

**S3 Fig. Top: Root-mean-square deviation (RMSD) of the protein backbone in each replicate.** In all three replicates the RMSD is still increasing at 500 ns, indicating that CmeC is not yet fully equilibrated. Bottom: When one replicate was extended to 1250 ns, the backbone RMSD continued to increase.
(TIF)

**S4 Fig. Secondary structure analysis (STRIDE) for CmeC from one replicate across the 500 ns simulation.**
(TIF)

**S5 Fig.** Timeseries data of the angle defined between the protein principal axis (approximately parallel to the z-axis) and the glycan vector (defined in Fig 4E). A wide variety of relative orientations are observed.
(TIF)

**S6 Fig. Distribution of end-to-end lengths for the heptasaccharide.** Ther is considerably more variation in this value than would be expected from a 'rigid-rod'.
(TIF)

**S7 Fig. Distribution of $\Psi$ and $\Phi$ values for each glycosidic linkage in the heptasaccharide.** Data from our equilibrium simulations plotted in blue. Values defined by the mean ± standard deviation in Ref 60 for each linkage shaded in yellow
(TIF)

**S8 Fig. Potassium ion density proximal to CmeC.** A small amount of potassium ion density is located near D86 within the channel.
(TIF)

**S9 Fig. Comparison of protein backbone RMSD from equilibrium simulations, systems where calcium ions were removed from within CmeC, and systems where both calcium and potassium ions were removed from within CmeC.** Systems where cations were removed saw increased RMSD values compared to equilibrium simulations, though this is most pronounced in systems where both calcium and potassium ions were removed.
(TIF)

**S10 Fig. Hydrogen bond count in the coiled-coil region of CmeC (hydrogen bonds between different helices only) for replicate 1 of the system with both calcium and potassium ions removed from within the channel.** Dashed pink vertical lines indicate times at which there was a peak in TCA[E416]; there is no clear increase or decrease in the number of these hydrogen bonds (inter- or intraprotomer) correlated with peaks in TCA.
(TIF)

**S11 Fig. Scatter plots of the force against the hydration shell population at positions within the CmeC channel, with points coloured by the channel radius, demonstrating correlation between the force required for translocation and both the channel radius and the population of the hydration shell.**
(TIF)

**S12 Fig. Occupancy map for a hydroxyl oxygen in Erythromycin (right, highlighted) over the steered MD simulations.** The occupancy is delocalised in the channel. The only clear localised region of high occupancy is at the periplasmic entrance, where the rotational freedom of erythromycin is limited due to the narrow channel diameter.
(TIF)

**S13 Fig. Electrostatic profile of erythromycin.** Colour scale: -1 V in red to +1 V in blue. Electronegative groups are distributed around the erythromycin molecule.
(TIF)

**S14 Fig. Electrostatic profile of CDCA.** Colour scale: -1 V in red to +1 V in blue. While there are three electronegative functional groups in CDCA, the carboxylate group dominates.
(TIF)

**S15 Fig. Occupancy map for a carboylate oxygen in chenodeoxycholic acid (right, highlighted) over the steered MD simulations.** There are several localised regions of high within the channel. These align with acidic lining residues where CDCA is coordinating a bound protein-bound calcium ion.
(TIF)

**S1 Table. Percentage variance explained by the first 10 eigenvectors identified via principal component analysis.**
(PDF)

**S2 Table. Mean values of $\Psi$ and $\Phi$ for each linkage in the heptasaccharide, with associated standard deviation.**
(PDF)

## Acknowledgments

The authors acknowledge access to the following High Performance Computing resources: Iridis 5 at the University of Southampton and the ARCHER2 UK National Computing Service to which access was granted via HECBioSim, the UK High-End Computing Consortium for Biomolecular Simulation (EPSRC grant no. EP/R029407/1).

## Author Contributions

**Conceptualization:** Syma Khalid.

**Data curation:** Kahlan E. Newman.

**Formal analysis:** Kahlan E. Newman.

**Funding acquisition:** Syma Khalid.

**Investigation:** Kahlan E. Newman.

**Methodology:** Kahlan E. Newman.

**Project administration:** Syma Khalid.

**Resources:** Syma Khalid.

**Software:** Kahlan E. Newman.

**Supervision:** Syma Khalid.

**Validation:** Kahlan E. Newman.

**Visualization:** Kahlan E. Newman.

**Writing – original draft:** Kahlan E. Newman.

**Writing – review & editing:** Kahlan E. Newman, Syma Khalid.

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
