## [Decision Letter · Decision Letter 0]

19 Oct 2022

Dear Professor KHALID,

Thank you very much for submitting your manuscript "Conformational dynamics and putative substrate extrusion pathways of the N-glycosylated outer membrane factor CmeC from Campylobacter jejuni" for consideration at PLOS Computational Biology. As with all papers reviewed by the journal, your manuscript was reviewed by members of the editorial board and by three independent reviewers. The reviewers appreciated the attention to an important topic. Based on the reviews, we are likely to accept this manuscript for publication, providing that you modify the manuscript according to the review recommendations.

Sincerely,

Guanghong Wei

Academic Editor

PLOS Computational Biology

Nir Ben-Tal

Section Editor

PLOS Computational Biology

Reviewer's Responses to Questions

**Comments to the Authors:**

Reviewer #1: In this manuscript, the authors conducted extensive molecular dynamics simulations to investigate the conformational dynamics of a N-glycosylated outer membrane factor CmeC and the possible extrusion pathway of small ligands which can be exported from the intracellular side to the extracellular side of the bilayer by this protein. The OMF is responsible for antibiotic resistance of the bacteria, Campylobacter jejuni, and therefore studying its mechanism of exporting small ligands is of interest to the community. The authors described in detail the conformation of the protein in equilibrium simulations, as well as how the saccharides affected its conformation. The authors also employed non-equilibrium simulations to pull two ligands across the channel. The authors concluded the manuscript with possible gates of the channel and a probably mechanism for negatively charged ligands using cations to climb a channel whose inner surface is full of acid residues.

Overall, the technique used in this work is solid, the results were analyzed properly and the manuscript was written in reasonable quality. I only have some minor issues which the authors may want to consider before the publication of this manuscript:

1, The authors used both potassium ions and calcium ions in their system. However, in their work, they mentioned that they found little potassium density inside the channel, but notable calcium density. What would be the possible reason for this difference? Is this a possible artifact of the used force field (as far as I know, the current ff with fixed charges are not perfect to describe ions, particularly multivalent cations), or there was some more fundamental reason? This should be discussed in the manuscript.

Also, the authors said that, they used calcium ions to neutralize the negatively charged LOS lipids. Why this particular cation is necessary for this purpose? Please clarify.

2, On the one hand, the authors mentioned that they identified higher calcium density than the potassium density inside the channel, on the other hand, the authors said that removing the calcium only increased the TCA slightly, while removing the both types of ions increased the TCA significantly. The reason should be explained. Why minor potassium had more dramatic effects than the more enriched calcium?

Probably a comparison of the distribution of two different ions in the channel would be helpful: this distribution will show if the potassium is really not distributed in the channel or if they are just not coordinate stably with the acid residues.

3, The authors performed principal component analyses in their work. The three simulation replicates showed complete different conformational space according to the principal components, what does this mean actually? It doesn’t seem to be a result of insufficient sampling.

Also, how the PCA analyses were conducted is not clear. Did the authors conducted PCA for each trajectory separately, or they combined all of the trajectories and then conducted PCA for the combined frames all together? If the former, did the corresponding principal components from three trajectories indicate the same collective motions? Please clarify.

The authors said the conformational space of three trajectories are unique from the crystal structure, but the PCs of the crystal structure was not labeled in Fig. 2. Besides, how much percentage did the first three PCs account for among all of the PCs (in another word, was it enough to only look at the first three PCs)?

4, the authors mentioned the two constriction regions at the periplasmic entrance and the extracellular exit in the introduction section, however, it is not clear where these regions are in Fig. 1. Of course, the positions were elaborated later in the manuscript, but I would suggest also label them in Fig. 1.

5, CDCA was supposed to utilize cations to climb the acid residue ladders of the channel, and this process was supposed to increase the ratio of the frames in which calcium ions were coordinated by acid residues. From Table 1, we can see that coordination of calcium by D77 was increased more significantly than the other residues, why this residue is so special? Please explain. For other residues, the increase was between 3%-10% (except for D86), did this minor difference has statistical significance?

6, the authors presented the orientation of the ligands in the channel by showing these ligands from different frames in one figure. However, I am afraid this might not be a perfect way to describe the ligand orientations. I suggest the authors to define other ways, such as define vectors and angles/dihedrals or show densities of specific atoms.

7, please clarify the concentration of calcium ions used in the work. The concentration of KCL was mentioned but not for the calcium ions.

Reviewer #2: This manuscript nicely describes the influence of structure and function for CmeC by lipid membrane models, ions, efflux of drugs. There is an extensive set of simulations that albeit not fully equilibrated provide some details on structural influences on CmeC by various conditions. Although I find this manuscript publishable, my comments and suggestions below should be addressed.

General Comments:

1. Outer membrane (OM) mimic: It is nice to see that the simulations utilize a mimic for the outer membrane with LOS, which better represents the natural environment for CmeC. However, the membrane model used is a mimic and not representative of the natural OM of C. jejuni. The authors should make this clear in the main document.

2. Equilibration (lack of): The authors are very clear and open about their research and that the protein is still changing in structure. It would be nice to provide a bit more detail on what regions of the protein are still changing. Moreover, does this relate to the crystal structure conditions vs. the more natural conditions simulated in this work? Some more physical insight into the inability to stabilize the structure would be good.

Reviewer #3: The manuscript entitled 'Conformational dynamics and putative substrate extrusion pathways of the N- glycosylated outer membrane factor CmeC from Campylobacter jejuni' by authors Kahlan Newman and Syma Khalid is an interesting study of the dynamics of a previously not dynamically investigated RND efflux pump outer duct. The work is technically sound in general, very well written, and provides interesting insights on the dynamics of loops, N-glycans and the trimer as a whole. Of particular interest is the role of the inner pore lining, which is especially rich in acidic side chains. The authors ought to be commended for using realistic outer membrane models and post-translational modification. I only have a few questions:

* The default model for calcium ion is known to overestimate protein-ion affinity (see for example doi: 10.1038/s41467-020-14573-w). The density and position of calcium ions near the protein is quite important for the results of this work - can the authors please comment why they chose the default model and discuss any possible implications for the results?

* I wonder why no attempt was made to estimate the free energy surface for substrate translocation by using the steered MD results along with the Jarzynski equality rather than by umbrella sampling since the steered MD trajectories already exist. Was there poor convergence for instance?

* Ion attraction sites, the electrostatic properties of the gating region, and pore gating in a similar outward efflux duct from Neisseria gonorrhoeae were studied by MD in doi: 10.1038/s41598-017-16995-x ; I think this paper is relevant previous work and the authors may want to discuss it.

* Are any inhibitors of this outward duct known and if so, where do they bind?

**Have the authors made all data and (if applicable) computational code underlying the findings in their manuscript fully available?**

Reviewer #1: Yes

Reviewer #2: Yes

Reviewer #3: Yes

PLOS authors have the option to publish the peer review history of their article (what does this mean?). If published, this will include your full peer review and any attached files.

Reviewer #1: No

Reviewer #2: No

Reviewer #3: No

Figure Files:

Data Requirements:

Reproducibility:

References:

---

## [Decision Letter · Decision Letter 1]

26 Dec 2022

Dear Professor KHALID,

We are pleased to inform you that your manuscript 'Conformational dynamics and putative substrate extrusion pathways of the N-glycosylated outer membrane factor CmeC from Campylobacter jejuni' has been provisionally accepted for publication in PLOS Computational Biology.

Best regards,

Guanghong Wei

Academic Editor

PLOS Computational Biology

Nir Ben-Tal

Section Editor

PLOS Computational Biology

Reviewer's Responses to Questions

**Comments to the Authors:**

Reviewer #1: All of my concerns have been addressed, I do not have more suggestions.

Reviewer #2: This revision is acceptable for publication.

Reviewer #3: all points have been addressed in style !

**Have the authors made all data and (if applicable) computational code underlying the findings in their manuscript fully available?**

Reviewer #1: Yes

Reviewer #2: Yes

Reviewer #3: None

PLOS authors have the option to publish the peer review history of their article (what does this mean?). If published, this will include your full peer review and any attached files.

Reviewer #1: No

Reviewer #2: No

Reviewer #3: No

---

## [Editor Report · Acceptance letter]

10 Jan 2023

PCOMPBIOL-D-22-01360R1 

Conformational dynamics and putative substrate extrusion pathways of the N-glycosylated outer membrane factor CmeC from Campylobacter jejuni

Dear Dr Khalid,

I am pleased to inform you that your manuscript has been formally accepted for publication in PLOS Computational Biology. Your manuscript is now with our production department and you will be notified of the publication date in due course.

With kind regards,

Zsofia Freund
